# Hyper-SD: Trajectory Segmented Consistency Model for Efficient Image Synthesis

**Yuxi Ren**   **Xin Xia**   **Yanzuo Lu**   **Jiacheng Zhang**   **Jie Wu**
**Pan Xie**   **Xing Wang**   **Xuefeng Xiao**[*]

ByteDance
Project Page: `https://hyper-sd.github.io/`

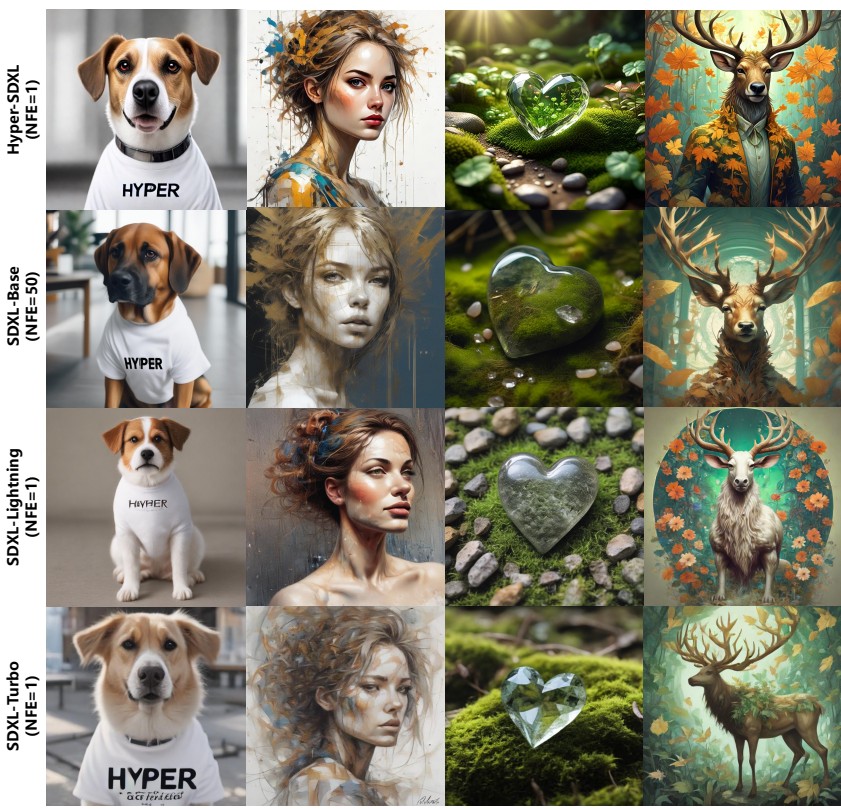

Figure 1. The visual comparison between our **Hyper-SDXL** and other methods. From the first to the fourth column, the prompts for these images are (1) a dog wearing a white t-shirt, with the word *"hyper"* written on it ... (2) *abstract beauty*, approaching perfection, pure form, golden ratio, minimalistic, unfinished,... (3) *a crystal heart* laying on moss in a serene zen garden ... (4) *anthropomorphic art of a scientist stag*, victorian inspired clothing by krenz cushart ...., respectively.

## Abstract

Recently, a series of diffusion-aware distillation algorithms have emerged to alleviate the computational overhead associated with the multi-step inference process of Diffusion Models (DMs). Current distillation techniques often dichotomize into two distinct aspects: i) ODE Trajectory Preservation; and ii) ODE Trajectory Reformulation. However, these approaches suffer from severe performance degradation or domain shifts. To address these limitations, we propose **Hyper-SD**, a

[*]Project Lead. Correspondence to <xiaoxuefeng.ailab@bytedance.com>.

38th Conference on Neural Information Processing Systems (NeurIPS 2024).

novel framework that synergistically amalgamates the advantages of ODE Trajectory Preservation and Reformulation, while maintaining near-lossless performance during step compression. Firstly, we introduce Trajectory Segmented Consistency Distillation to progressively perform consistent distillation within pre-defined time-step segments, which facilitates the preservation of the original ODE trajectory from a higher-order perspective. Secondly, we incorporate human feedback learning to boost the performance of the model in a low-step regime and mitigate the performance loss incurred by the distillation process. Thirdly, we integrate score distillation to further improve the low-step generation capability of the model and offer the first attempt to leverage a unified LoRA to support the inference process at all steps. Extensive experiments and user studies demonstrate that Hyper-SD achieves SOTA performance from 1 to 8 inference steps for both SDXL and SD1.5. For example, Hyper-SDXL surpasses SDXL-Lightning by *+0.68* in CLIP Score and *+0.51* in Aes Score in the 1-step inference.

# 1   Introduction

Diffusion models (DMs) have gained significant prominence in the field of Generative AI [9, 25, 20, 24], but they are burdened by the computational requirements[36, 12] associated with multi-step inference procedures [27, 10]. To overcome these challenges and fully exploit the capabilities of DMs, several distillation methods have been proposed [27, 32, 46, 10, 16, 29, 14, 40, 28], which can be categorized into two main groups: trajectory-preserving distillation and trajectory-reformulating distillation.

Trajectory-preserving distillation techniques are designed to maintain the original trajectory of an ordinary differential equation (ODE) [27, 46]. The primary objective of these methods is to enable student models to make further predictions on the flow and reduce the overall number of inference steps. These techniques prioritize the preservation of similarity between the outputs of the distilled model and the original model. Adversarial losses can also be employed to enhance the accuracy of supervised guidance in the distillation process [14]. However, it is important to note that, despite their benefits, trajectory-preserved distillation approaches may suffer from a decrease in generation quality due to inevitable errors in model fitting.

Trajectory-reformulating methods directly utilize the endpoint of the ODE flow or real images as the primary source of supervision, disregarding the intermediate steps of the trajectory [16, 29, 28]. By reconstructing more efficient trajectories, these methods can also reduce the number of inference steps. Trajectory-reformulating approaches enable the exploration of the model's potential within a limited number of steps, liberating it from the constraints of the original trajectory. However, it can lead to inconsistencies between the accelerated model and the original model's output domain, often resulting in undesired effects.

To navigate these hurdles and harness the full potential of DMs, we present an advanced framework that adeptly combines trajectory-preserving and trajectory-reformulating distillation techniques. Firstly, we proposed trajectory segmented consistency distillation (TSCD), which divides the time steps into segments and enforces consistency within each segment while gradually reducing the number of segments to achieve all-time consistency. This approach addresses the issue of suboptimal consistency model performance caused by insufficient model fitting capability and accumulated errors in inference. Secondly, we leverage human feedback learning techniques [37, 44, 23] to optimize the accelerated model, modifying the ODE trajectories to better suit few-step inference. This results in significant performance improvements, even surpassing the capabilities of the original model in some scenarios. Thirdly, we enhanced the one-step generation performance using score distillation [35, 40], achieving the idealized all-time consistent model via a unified LoRA.

In summary, our main contributions are summarized as follows: **1) Accelerate**: we propose TSCD that achieves a more fine-grained and high-order consistency distillation approach for the original score-based model. **2) Boost**: we incorpoate human feedback learning to further enhance model performance in low-steps regime. **3) Unify**: we provide a unified LORA as the all-time consistency model and support inference at all NTEs. **4) Performance**: Hyper-SD achieves SOTA performance in low-steps inference for both SDXL and SD1.5.

## 2 Preliminaries

For completeness, the preliminaries on diffusion model are provided in Appendix A.

### 2.1 Diffusion Model Distillation

As mentioned in Sec. 1, current techniques for distilling Diffusion Models (DMs) can be broadly categorized into two approaches: one that preserves the Ordinary Differential Equation (ODE) trajectory [27, 32, 46, 10], and another that reformulates it [29, 14, 40, 28].

Here, we provide a concise overview of some representative categories of methods. For clarity, we define the teacher model as $f_{tea}$, the student model as $f_{stu}$, noise as $\epsilon$, prompt condition as $c$, off-the-shelf ODE Solver as $\Psi(\cdot, \cdot, \cdot)$, the total training timesteps as $T$, the num of inference timesteps as $N$, the noised trajectory point as $x_t$ and the skipping-step as $s$, where $t_0 < t_1 \cdots < t_{N-1} = T$, $t_n - t_{n-1} = s$, $n$ uniformly distributed over $\{1, 2, \ldots, N-1\}$.

**Progressive Distillation.** Progressive Distillation (PD) [27] trains the student model $f_{stu}$ approximate the subsequent flow locations determined by the teacher model $f_{tea}$ over a sequence of steps.

Considering a 2-step PD for illustration, the target prediction $\hat{x}_{t_{n-2}}$ by $f_{tea}$ is obtained through the following calculations:

$$\hat{x}_{t_{n-1}} = \Psi(x_{t_n}, f_{tea}(x_{t_n}, t_n, c), t_{n-1}), \tag{1}$$

$$\hat{x}_{t_{n-2}} = \Psi(\hat{x}_{t_{n-1}}, f_{tea}(\hat{x}_{t_{n-1}}, t_{n-1}, c), t_{n-2}), \tag{2}$$

And the training loss is

$$\mathcal{L}_{PD} = \|\hat{x}_{t_{n-2}} - \Psi(x_{t_n}, f_{stu}(x_{t_n}, t_n, c), t_{n-2})\|_2^2 \tag{3}$$

**Consistency Distillation.** Consistency Distillation (CD) [32] directly maps $x_{t_n}$ along the ODE trajectory to its endpoint $x_0$. The training loss is defined as :

$$\mathcal{L}_{CD} = \|\Psi(x_{t_n}, f_{stu}(x_{t_n}, t_n, c), 0) - \Psi(\hat{x}_{t_{n-1}}, f_{stu}^-(\hat{x}_{t_{n-1}}, t_{n-1}, c), 0)\|_2^2 \tag{4}$$

where $f_{stu}^-$ is the exponential moving average(EMA) of $f_{stu}$ and $\hat{x}_{t_{n-1}}$ is the next flow location estimated by $f_{tea}$ with the same function as Eq. (1).

The Consistency Trajectory Model (CTM) [10] was introduced to minimize accumulated estimation errors and discretization inaccuracies prevalent in multi-step consistency model sampling. Diverging from targeting the endpoint $x_0$, CTM targets any intermediate point $x_{t_{end}}$ within the range $0 \leq t_{end} \leq t_{n-1}$, thus redefining the loss function as:

$$\mathcal{L}_{CTM} = \|\Psi(x_{t_n}, f_{stu}(x_{t_n}, t_n, c), t_{end}) - \Psi(\hat{x}_{t_{n-1}}, f_{stu}^-(\hat{x}_{t_{n-1}}, t_{n-1}, c), t_{end})\|_2^2 \tag{5}$$

**Adversarial Diffusion Distillation.** In contrast to PD and CD, Adversarial Distillation (ADD), proposed in SDXL-Turbo [29] and SD3-Turbo [28], bypasses the ODE trajectory and directly focuses on the original state $x_0$ using adversarial objective. The generative and discriminative loss components are computed as follows:

$$\mathcal{L}_{ADD}^G = -\mathbb{E}\left[D(\Psi(x_{t_n}, f_{stu}(x_{t_n}, t_n, c), 0))\right] \tag{6}$$

$$\mathcal{L}_{ADD}^D = \mathbb{E}\left[D(\Psi(x_{t_n}, f_{stu}(x_{t_n}, t_n, c), 0))\right] - \mathbb{E}\left[D(x_0)\right] \tag{7}$$

where $D$ denotes the discriminator, tasked with differentiating between $x_0$ and $\Psi(x_{t_n}, f_{stu}(x_{t_n}, t_n, c), 0)$. The target $x_0$ can be sampled from real or synthesized data.

**Score Distillation Sampling.** Score distillation sampling(SDS)[21] was integrated into diffusion distillation in SDXL-Turbo[29] and Diffusion Matching Distillation(DMD)[40]. SDXL-Turbo[29] utilizes $f_{tea}$ to estimate the score to the real distribution, while DMD[40] further introduced a fake distribution simulator $f_{fake}$ to calibrate the score direction and uses the output distribution of the original model as the real distribution, thus achieving one-step inference.

Leveraging the DMD approach, the gradient of the Kullback-Leibler (KL) divergence between the real and fake distributions is approximated by the equation:

$$\nabla D_{KL} = \mathop{\mathbb{E}}_{\substack{z \sim \mathcal{N}(0, I) \\ x = f_{stu}(z)}} \left[-(f_{real}(x) - f_{fake}(x))\nabla f_{stu}(z)\right], \tag{8}$$

where $z$ is a random latent variable sampled from a standard normal distribution. This methodology enables the one-step diffusion model to refine its generative process, minimizing the KL divergence to produce images that are progressively closer to the teacher model's distribution.

## 2.2 Human Feedback Learning

ReFL [37, 13, 44] has been proven to be an effective method to learn from human feedback designed for diffusion models. It primarily includes two stages: (1) reward model training and (2) preference fine-tuning. In the first stage, given the human preference data pair, $x_w$ (preferred generation) and $x_l$ (unpreferred one), a reward model $r_\theta$ is trained via the loss:

$$\mathcal{L}(\theta)_{rm} = -\mathbb{E}_{(c,x_w,x_l)\sim\mathcal{D}}[log(\sigma(r_\theta(c, x_w) - r_\theta(c, x_l)))] \tag{9}$$

where $\mathcal{D}$ denotes the collected feedback data, $\sigma(\cdot)$ represents the sigmoid function, and $c$ corresponds to the text prompt. The reward model $r_\theta$ is optimized to produce reward scores that align with human preferences. In the second stage, ReFL starts with an input prompt $c$, and a randomly initialized latent $x_T = z$. The latent is then iteratively denoised until reaching a randomly selected timestep $t_n \in [t_{left}, t_{right}]$, when a denoised image $x'_0$ is directly predicted from $x_{t_n}$. The $t_{left}$ and $t_{right}$ are predefined boundaries. The reward model is then applied to this denoised image, generating the expected preference score $r_\theta(c, x'_0)$, which is used to fine-tuned the diffusion model:

$$\mathcal{L}(\theta)_{refl} = \mathbb{E}_{c\sim p(c)}\mathbb{E}_{x'_0\sim p(x'_0|c)}[-r(x'_0, c)] \tag{10}$$

## 3 Method

In this study, we have integrated both the ODE-preserve and ODE-reformulate distillation techniques into a unified framework, yielding significant advancements in accelerating diffusion models. In Sec. 3.1, we propose an innovative approach to consistency distillation that employs a time-steps segmentation strategy, thereby facilitating trajectory segmented consistency distillation. In Sec. 3.2, we incorporate human feedback learning techniques to further enhance the performance of accelerated diffusion models. In Sec. 3.3, we achieve all-time consistency including one-step by utilizing the score-based distribution matching distillation.

## 3.1 Trajectory Segmented Consistency Distillation

Both Consistency Distillation (CD) [32] and Consistency Trajectory Model (CTM) [10] aim to transform a diffusion model into a consistency model across the entire timestep range $[0, T]$ through single-stage distillation. However, these distilled models often fall short of optimality due to limitations in model fitting capacity. Drawing inspiration from the soft consistency target introduced in CTM, we refine the training process by dividing the entire time-steps range $[0, T]$ into $k$ segments and performing segment-wise consistent model distillation progressively.

In the first stage, we set $k = 8$ and use the original diffusion model to initiate $f_{stu}$ and $f_{tea}$. The starting timesteps $t_n$ are uniformly and randomly sampled from $\{t_1, t_2, \ldots, t_{N-1}\}$. We then sample ending timesteps $t_{end} \in [t_b, t_{n-1}]$, where $t_b$ is computed as:

$$t_b = \left\lfloor \frac{t_n}{\left\lfloor \frac{T}{k} \right\rfloor} \right\rfloor \times \left\lfloor \frac{T}{k} \right\rfloor, \tag{11}$$

and the training loss is calculated as:

$$L_{TSCD} = d(\Psi(x_{t_n}, f_{stu}(x_{t_n}, t_n, c), t_{end}), \Psi(\hat{x}_{t_{n-1}}, f^-_{stu}(\hat{x}_{t_{n-1}}, t_{n-1}, c), t_{end})) \tag{12}$$

where $\hat{x}_{t_{n-1}}$ refers to Eq. (1), and $f^-_{stu}$ denotes the Exponential Moving Average (EMA) of $f_{stu}$.

Subsequently, we resume the model weights from the previous stage and continue to train $f_{stu}$, progressively reducing $k$ to $[4, 2, 1]$. It is noteworthy that $k = 1$ corresponds to the standard CTM training protocol. For the distance metric $d$, we employ a hybrid of adversarial loss, as proposed in sdxl-lightning[14], and Mean Squared Error (MSE) Loss. Empirically, we observe that MSE Loss is more effective when the predictions and target values are proximate (e.g., for $k = 8, 4$), whereas adversarial loss proves more precise as the divergence between predictions and targets increases (e.g., for $k = 2, 1$). Accordingly, we dynamically increase the weight of the adversarial loss and diminish that of the MSE loss across the training stages. Additionally, we have integrated a noise perturbation mechanism [8] to reinforce training stability. Take the two-stage Trajectory Segmented Consistency Distillation (TSCD) process as an example. As shown in Fig. 2, the first stage executes

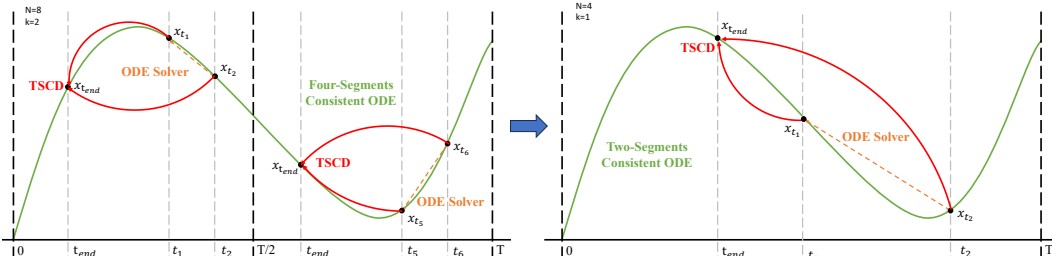

Figure 2. An illustration of the two-stage Trajectory Segmented Consistency Distillation. The first stage involves consistency distillation in two separate time segments: $[0, \frac{T}{2}]$ and $[\frac{T}{2}, T]$ to obtain the two segments consistency ODE. Then, this ODE trajectory is adopted to train a global consistency model in the subsequent stage.

independent consistency distillations within the time segments $[0, \frac{T}{2}]$ and $[\frac{T}{2}, T]$. Based on the previous two-segment consistency distillation results, a global consistency trajectory distillation is then performed.

The TSCD method offers two principal advantages: Firstly, fine-grained segment distillation reduces model fitting complexity and minimizes errors, thus mitigating degradation in generation quality. Secondly, it ensures the preservation of the original ODE trajectory. Models from each training stage can be utilized for inference at corresponding steps while closely mirroring the original model's generation quality. We illustrate the complete procedure of Trajectory Segmented Consistency Distillation in Appendix B. It is worth noting that, by utilizing Low-Rank Adaptation(LoRA) technology, we train TSCD models as plug-and-play plugins that can be used instantly.

## 3.2 Human Feedback Learning

In addition to the distillation, we propose to incorporate feedback learning further to boost the performance of the accelerated diffusion models. In particular, we improve the generation quality of the accelerated models by exploiting the feedback drawn from both human aesthetic preferences and existing visual perceptual models. For the feedback on aesthetics, we utilize the LAION aesthetic predictor and the aesthetic preference reward model provided by ImageReward[37] to steer the model toward the higher aesthetic generation as:

$$\mathcal{L}(\theta)_{aes} = \sum \mathbb{E}_{c \sim p(c)} \mathbb{E}_{x'_0 \sim p(x'_0|c)} \big[ \texttt{ReLU}(\alpha_d - r_d(x'_0, c)) \big] \tag{13}$$

where $r_d$ is the aesthetic reward model, including the aesthetic predictor of the LAION dataset and ImageReward model, $c$ is the textual prompt and $\alpha_d$ together with ReLU function works as a hinge loss.

Beyond the feedback from aesthetic preference, we notice that the existing visual perceptual model embedded in rich prior knowledge about the reasonable image can also serve as a good feedback provider. Empirically, we found that the instance segmentation model can guide the model to generate entities with reasonable structure. To be specific, instead of starting from a random initialized latent, we first diffuse the noise on an image $x_0$ in the latent space to $x_t$ according to Eq. (16), and then, we execute denoise iteratively until a specific timestep $d_t$ and directly predict a $x'_0$ similar to [37]. Subsequently, we leverage perceptual instance segmentation models to evaluate the performance of structure generation by examining the perceptual discrepancies between the ground truth image instance annotation and the predicted results on the denoised image as:

$$\mathcal{L}(\theta)_{percep} = \mathop{\mathbb{E}}_{\substack{x_0 \sim \mathcal{D} \\ x'_0 \sim G(x_{t_a})}} \mathcal{L}_{instance}((m_I(x'_0)), GT(x_0)) \tag{14}$$

where $m_I$ is the instance segmentation model(e.g. SOLO [34]). The instance segmentation model can capture the structure defect of the generated image more accurately and provide a more targeted feedback signal. It is noteworthy that besides the instance segmentation model, other perceptual models are also applicable and we are actively investigating the utilization of advanced large visual perception models(e.g. SAM) to provide enhanced feedback learning. Such perceptual models can work as complementary feedback for the subjective aesthetic focusing more on the objective

generation quality. Therefore, we optimize the diffusion models with the feedback signal as:

$$\mathcal{L}(\theta)_{feedback} = \mathcal{L}(\theta)_{aes} + \mathcal{L}(\theta)_{percep} \qquad (15)$$

Human feedback learning can improve model performance but may unintentionally alter the output domain, which is not always desirable. Therefore, we also trained human feedback learning knowledge as a plugin using LoRA technology. By employing the LoRA merge technique with the TSCD LoRAs discussed in Section3.1, we can achieve a flexible balance between generation quality and output domain similarity.

### 3.3 One-step Generation Enhancement

One-step generation within the consistency model framework is not ideal due to the inherent limitations of consistency loss. As analyzed in Fig. 3, the consistency distilled model demonstrates superior accuracy in guiding towards the trajectory endpoint $x_0$ at position $x_t$. Therefore, score distillation is a suitable and efficient way to boost the one-step generation of our TSCD models.

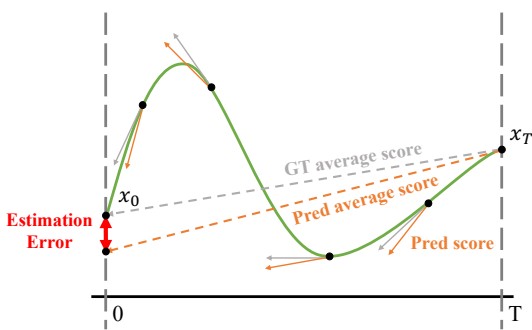

Figure 3. Score distillation comparison between score-based model and consistency model. The estimated score produced by the score-based model may exhibit a greater estimation error than the consistency model.

Specifically, we advance one-step generation with an optimized Distribution Matching Distillation (DMD) technique [40]. DMD enhances the model's output by leveraging two distinct score functions: $f_{real}(x)$ from the teacher model's distribution and $f_{fake}(x)$ from the fake model. We incorporate a Mean Squared Error (MSE) loss alongside the score-based distillation to promote training stability. The human feedback learning technique mentioned in Sec. 3.2 is also integrated, fine-tuning our models to efficiently produce images of exceptional fidelity.

After enhancing the one-step inference capability of the TSCD model, we can obtain an ideal global consistency model. Employing the TCD scheduler[46], the enhanced model can perform inference from 1 to 8 steps. Our approach eliminates the need for model conversion to x0-prediction[14], enabling the implementation of the one-step LoRA plugin. We demonstrated the effectiveness of our **one-step LoRA** in Sec 4.3. Additionally, smaller time-step inputs can enhance the credibility of the one-step diffusion model in predicting the noise [7]. Therefore, we also employed this technique to train a dedicated model for single-step generation.

## 4 Experiments

### 4.1 Implementation Details

**Dataset.** We use a subset of the LAION [30] and COYO [6] datasets following SDXL-lightning [14] during the training procedure of Sec 3.1 and Sec 3.3. For the Human Feedback Learning in Sec 3.2, we utilized the COCO2017 train

Table 1. Comparison with other acceleration approaches.

| Method | Step | Support Arch. | CFG Free | One-Step UNet | One-Step LoRA |
|---|---|---|---|---|---|
| PeRFlow [38] | 4+ | SD15 | No | No | No |
| TCD [46] | 2+ | SD15/XL | Yes | No | No |
| LCM [19] | 2+ | SD15/XL | Yes | No | No |
| Turbo [29] | 1+ | SD21/XL | Yes | Yes | No |
| Lightning [14] | 1+ | SDXL | Yes | Yes | No |
| **Ours** | **1+** | **SD15/XL** | **Yes** | **Yes** | **Yes** |

split dataset with instance annotations and captions for structure optimization.

**Training Setting.** For TSCD in Sec 3.1, we progressively reduced the time-steps segments number as $8 \to 4 \to 2 \to 1$ in four stages, employing 512 batch size and learning rate $1e-6$. We take the SOLO [34] as the instance segmentation model to achieve feedback learning in Sec 3.2. Our training per stage costs around 200 A100 GPU hours. We trained LoRA instead of UNet for all the distillation stages for convenience, and the corresponding LoRA is loaded to process the human feedback learning optimization in Sec 3.2. For one-step enhancement in Sec 3.3, we trained the unified all-timesteps consistency LoRA with time-step inputs $T = 999$ and the dedicated model for single-step generation with $T = 800$.

**Baseline Models.** We conduct our experiments on the stable-diffusion-v1-5 (SD15) [25] and stable-diffusion-xl-v1.0-base (SDXL) [20]. To demonstrate the superiority of our method in acceleration, we compared our method with various existing acceleration schemes as shown in Tab. 1.

**Evaluation Metrics.** We use the aesthetic predictor pre-trained on the LAION dataset and CLIP score(ViT-B/32) to evaluate the visual appeal of the generated image and the text-to-image alignment. We further include some recently proposed metrics, such as ImageReward score [37], and Pickscore [11] to offer a more comprehensive evaluation of the model performance. In addition to these, due to the inherently subjective nature of image generation evaluation, we conduct an extensive user study to evaluate the performance more accurately.

## 4.2 Main Results

**Quantitative Comparison.** We quantitatively compare our method with both the baseline and diffusion-based distillation approaches in terms of objective metrics. The evaluation is performed on COCO-5k [15] dataset with both SD15 (512px) and SDXL (1024px) architectures. As shown in Tab. 2, our method significantly outperforms the state-of-the-art across all metrics on both resolutions. In particular, compared to the two baseline models, we achieve better aesthetics (including AesScore, ImageReward and PickScore) with only LoRA and fewer steps. As for the CLIPScore

Table 2. Quantitative comparisons with state-of-the-arts on SD15 and SDXL architectures. The best result is highlighted in **bold**.

| Model | Step | Type | CLIP Score | Aes Score | Image Reward | Pick Score |
|---|---|---|---|---|---|---|
| SD15-Base [25] | 25 | UNet | 31.88 | 5.26 | 0.18 | 0.217 |
| SD15-LCM [19] | 4 | LoRA | 30.36 | 5.66 | -0.37 | 0.212 |
| SD15-TCD [46] | 4 | LoRA | 30.62 | 5.45 | -0.15 | 0.214 |
| PeRFlow [38] | 4 | UNet | 30.77 | 5.64 | -0.35 | 0.208 |
| **Hyper-SD15** | 1 | LoRA | **30.87** | **5.79** | **0.29** | **0.215** |
| SDXL-Base [25] | 25 | UNet | 33.16 | 5.54 | 0.87 | 0.229 |
| SDXL-LCM [19] | 4 | LoRA | 32.43 | 5.42 | 0.48 | 0.224 |
| SDXL-TCD [46] | 4 | LoRA | 32.45 | 5.42 | 0.67 | 0.226 |
| SDXL-Lightning [14] | 4 | LoRA | 32.40 | 5.63 | 0.72 | 0.229 |
| **Hyper-SDXL** | 4 | LoRA | **32.56** | **5.74** | **0.93** | **0.232** |
| SDXL-Turbo [29] | 1 | UNet | 32.33 | 5.33 | 0.78 | 0.228 |
| SDXL-Lightning [14] | 1 | UNet | 32.17 | 5.34 | 0.54 | 0.223 |
| **Hyper-SDXL** | 1 | UNet | **32.85** | **5.85** | **1.19** | **0.231** |

that evaluates image-text matching, we outperform other methods by +0.1 faithfully and are also closest to the baseline model, which demonstrates the effectiveness of our human feedback learning.

**Qualitative Comparison.** We present comprehensive visual comparison with recent approaches, including LCM [19], TCD [46], PeRFLow [38], Turbo [29] and Lightning [14]. Our observations can be summarized as follows. **(1)** Thanks to the fact that SDXL has almost 2.6B parameters, the model is able to synthesis decent images in 4 steps after different distillation algorithms. Our method further utilizes its huge model capacity to compress the number of steps required for high-quality outcomes to 1 step only, and far outperforms other methods in terms of style (a), aesthetics (b-c) and image-text matching (d) as indicated in Fig. 4. **(2)** On the contrary, limited by the capacity of SD15 model, the images generated by other approaches tend to exhibit severe quality degradation. While our Hyper-SD consistently yields better results across different types of user prompts, including photographic (a), realistic (b-c) and artstyles (d) as depicted in Appendix C.1. **(3)** To further release the potential of our methodology, we also conduct experiments on the fully fine-tuning of SDXL model following previous works [14, 29]. As shown in Appendix C.2, our 1-Step UNet again demonstrates superior generation quality that far exceeds the rest of the opponents. Both in terms of colorization (a-b) and details (c-d), our images are more presentable and attractive when it comes to the real-world application scenarios.

**User Study.** To verify the effectiveness of our proposed Hyper-SD, we conduct an extensive user study across various settings and approaches. As presented in Fig. 5, our method (red in left) obtains significantly more user preferences than others (blue in right). Specifically, our Hyper-SD15 has achieved more than a two-thirds advantage against the same architectures. The only exception is that SD21-Turbo [22] was able to get significantly closer to our generation quality in one-step inference by means of a larger training dataset of SD21 model as well as fully fine-tuning. Notably, we found that we obtained a higher preference with less inference steps compared to both the baseline SD15 and SDXL models, which once again confirms the validity of our human feedback learning. Moreover, our 1-Step UNet shows a higher preference than LoRA against the same UNet-based approaches (i.e. SDXL-Turbo [29] and SDXL-Lightning [14]), which is also consistent with the analyses of previous quantitative and qualitative comparisons. This demonstrates the excellent scalability of our method when more parameters are fine-tuned.

| SDXL-Base | SDXL-LCM | SDXL-TCD | SDXL-Lightning | Hyper-SDXL | |
|---|---|---|---|---|---|
| 50NFE, CFG7.5 | No CFG | No CFG | No CFG | No CFG | |
| 25 Steps | 4 Steps | 4 Steps | 4 Steps | 1 Step (LoRA) | 4 Steps |

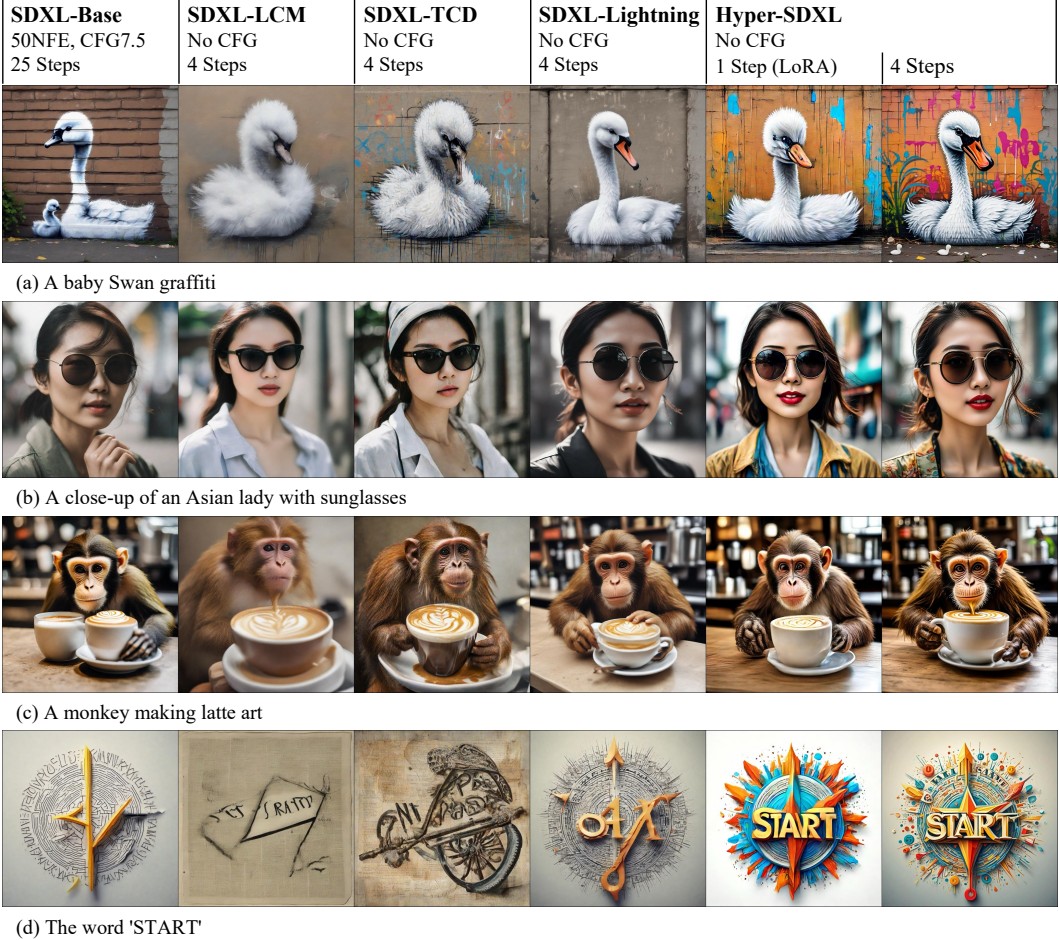

(a) A baby Swan graffiti

(b) A close-up of an Asian lady with sunglasses

(c) A monkey making latte art

(d) The word 'START'

Figure 4. Qualitative comparisons with LoRA-based approaches on SDXL architecture.

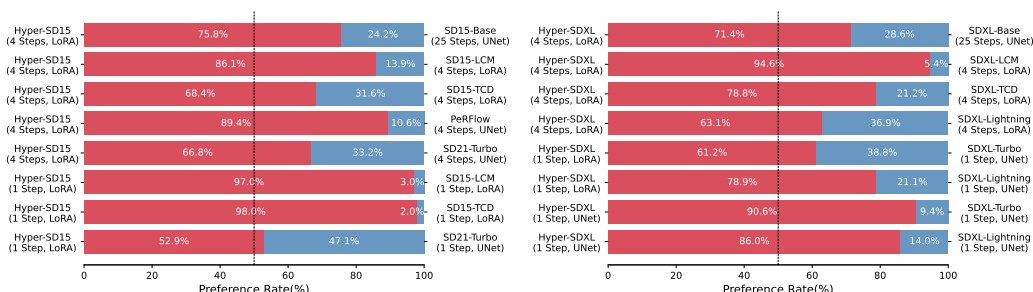

Figure 5. The user study about the comparison between our method and other methods.

## 4.3 Ablation Study

In Tab. 3, we provide ablation studies on TSCD and human feedback with quantitative evaluation.

**Effect of TSCD.** Without the utilization of human feedback, the results show that our proposed TSCD outperforms the baseline TCD [46] significantly when inference step is lower, while the performance approaches the same as the step increases. This demonstrates the effectiveness of our progressive strategy that alleviates the training difficulties when the step is extremely low.

**Effect of Human Feedback.** To verify the benefit of incorporating reward models, we also conduct experiments ablating on human feedback learning of different steps. As shown in Tab. 3, the

performance degradation caused by distillation process is well compensated after human feedback learning. Moreover, the image-text alignment (i.e. CLIPScore) and aesthetics (i.e. Others) evaluated on different steps are very similar, which better matches the definition of consistency model [32].

**Effect of One-Step Enhancement.** In Tab. 3, we also re-implement the DMD [40] with human feedback. The results show that our TSCD model indeed exhibit less estimation error than the original score-based model with similar text-to-image alignment and better aesthetic metrics, demonstrating higher generation quality and utility.

**Unified LoRA.** In addition to the different steps of LoRAs proposed above, we note that our one-step LoRA can be considered as a unified approach, since it is able to reason about different number of steps (e.g. 1,2,4,8 as shown in Appendix C.3) and consistently generate high-quality results under the effect of consistency distillation. For completeness, Tab. 4 also presents the quantitative results of different steps when applying the 1-Step unified LoRA. We can observe that there is no difference in image-text matching between different steps as the CLIPScore evaluates, which means that user prompts are well adhered to. And as the other metrics show, the aesthetics rise slightly as the step increases, which is as expected after all the user can choose based on the needs for efficiency. This would be of great convenience and practicality in real-world deployment scenarios, since generally only one model can be loaded per instance.

Table 3. Ablation studies of TSCD and human feedback.

| Method | Step | CLIP Score | Aes Score | Image Reward | Pick Score |
|---|---|---|---|---|---|
| *SDXL Architecture (w/o Human Feedback)* | | | | | |
| TCD [46] | 2 | 32.36 | 5.62 | 0.29 | 0.217 |
| **TSCD (Ours)** | 2 | 32.49 | 5.58 | 0.64 | 0.222 |
| TCD [46] | 4 | 32.45 | 5.42 | 0.67 | 0.226 |
| **TSCD (Ours)** | 4 | 32.53 | 5.66 | 0.78 | 0.229 |
| TCD [46] | 8 | 32.47 | 5.78 | 0.76 | 0.229 |
| **TSCD (Ours)** | 8 | 32.46 | 5.85 | 0.77 | 0.229 |
| *SDXL Architecture (w/ Human Feedback)* | | | | | |
| DMD [40] | 1 | 32.70 | 5.58 | 0.82 | 0.223 |
| **TSCD (Ours)** | 1 | 32.59 | 5.69 | 1.06 | 0.226 |
| **TSCD (Ours)** | 2 | 32.61 | 5.84 | 1.04 | 0.232 |
| **TSCD (Ours)** | 4 | 32.56 | 5.74 | 0.93 | 0.232 |
| **TSCD (Ours)** | 8 | 32.56 | 5.89 | 0.93 | 0.232 |

Table 4. Quantitative results on unified LoRAs.

| Arch. | Step | CLIP Score | Aes Score | Image Reward | Pick Score |
|---|---|---|---|---|---|
| **SD15** 512px | 8 | 30.73 | 5.47 | 0.53 | 0.224 |
| | 4 | 31.07 | 5.55 | 0.53 | 0.224 |
| | 2 | 31.21 | 5.93 | 0.45 | 0.222 |
| | 1 | 30.87 | 5.79 | 0.29 | 0.215 |
| **SDXL** 1024px | 8 | 32.54 | 5.83 | 1.14 | 0.233 |
| | 4 | 32.51 | 5.52 | 1.15 | 0.234 |
| | 2 | 32.59 | 5.71 | 1.15 | 0.234 |
| | 1 | 32.59 | 5.69 | 1.06 | 0.226 |

**Compatibility with ControlNet.** Appendix C.4 shows that our models are also compatible with ControlNet [45]. We test the one-step unified SD15 and SDXL LoRAs on the scribble [4] and canny [1] control images, respectively. And we can observe the conditions are well followed and the consistency of our unified LoRAs can still be demonstrated, where the quality of generated images under different inference steps are always guaranteed.

**Compatibility with Base Model.** Appendix C.5 shows that our LoRAs can be applied to different base models. Specifically, we conduct experiments on anime [2], realistic [3] and artstyle [5] base models. The results demonstrate that our method has a wide range of applications, and the lightweight LoRA also significantly reduces the cost of acceleration.

## 5   Conclusion

We propose Hyper-SD, a unified framework that maximizes the few-step generation capacity of diffusion models, achieving new SOTA performance based on SDXL and SD15. By employing trajectory-segmented consistency distillation, we enhanced the trajectory preservation ability during distillation, approaching the generation proficiency of the original model. Then, human feedback learning and variational score distillation stimulated the potential for few-step inference, resulting in a more optimal and efficient trajectory for generating models. We have open-sourced LoRAs for SDXL and SD15 from 1 to 8 steps inference, along with a dedicated one-step SDXL model, aiming to further propel the development of generative AI community. More discussions refer to Appendix E.

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

# A    Preliminaries of Diffusion Model

Diffusion models (DMs), as introduced by Ho et al. [9], consist of a forward diffusion process, described by a stochastic differential equation (SDE) [33], and a reverse denoising process. The forward process gradually adds noise to the data, transforming the data distribution $p_{\text{data}}(x)$ into a known distribution, typically Gaussian. This process is described by:

$$\mathrm{d}x_t = \mu(x_t, t)\mathrm{d}t + \sigma(t)\mathrm{d}w_t, \tag{16}$$

where $t \in [0, T]$, $w_t$ represents the standard Brownian motion, $\mu(\cdot, \cdot)$ and $\sigma(\cdot)$ are the drift and diffusion coefficients respectively. The distribution of $x_t$ sampled during the diffusion process is denoted as $p_t(x)$, with the empirical data distribution $p_0(x) \equiv p_{\text{data}}(x)$, and $p_T(x)$ being approximated by a tractable Gaussian distribution.

This SDE is proved to have the same solution trajectories as an ordinary differential equation (ODE) [33], dubbed as Probability Flow (PF) ODE, which is formulated as

$$\mathrm{d}x_t = \left[\mu(x_t, t) - \frac{1}{2}\sigma(t)^2 \nabla_{x_t} \log p_t(x_t)\right] \mathrm{d}t. \tag{17}$$

Therefore, the DM $s_\theta(x, t)$ is trained to estimate the score function $\nabla_{x_t} \log p_t(x_t)$. Then the estimation can be used to approximate the above PF ODE by an empirical PF ODE. Although various efficient methods [27, 32, 46, 10, 16, 29, 14, 40, 28] have been proposed to solve the ODE, the quality of the generated images $x_0$ is still not optimal when using relatively large $\mathrm{d}t$ steps. This underlines the necessity for multi-step inference in DMs and presents a substantial challenge to their wider application. For example, several customized diffusion models [26, 39? ] still require 50 inference steps to generate high-quality images although the overhead has been greatly reduced during training.

# B    Pseudo Code of TSCD

---

**Algorithm 1** Trajectory Segmented Consistency Distillation (TSCD)

---

1: **Input:** dataset $\mathcal{D}$, initial model parameters $\Theta$, learning rate $\eta$, ODE solver $\Psi$, noise schedule functions $\alpha(t)$ and $\sigma(t)$, guidance scale range $[\omega_{\min}, \omega_{\max}]$, the total segment count list $k_{\text{List}}$, the skipping-step as $s$, total training timesteps $T$, the num of inference timesteps list $N_{\text{List}}$ and encoder function $E(\cdot)$.
2: **Initialize:** Set the EMA of model parameters $\Theta^- \leftarrow \Theta$.
3: **for** $(i, k)$ in enumerate($k_{\text{List}}$) **do**
4:     Compute the num of inference timesteps $N = N_{\text{List}}[i]$
5:     **for** each training iteration **do**
6:         Sample batch $(x, c)$ from dataset $\mathcal{D}$, and guidance scale $\omega$ from $U[\omega_{\min}, \omega_{\max}]$.
7:         Compute the training timesteps $\{t_0, t_1, \ldots, t_{N-1}\}$ such that $t_0 < t_1 < \cdots < t_{N-1} = T$ with a uniform step size $s$, where $t_n - t_{n-1} = s$ for $n$ uniformly distributed over $\{1, 2, \ldots, N - 1\}$.
8:         Sample starting timestep $t_n$ uniformly from $\{t_1, t_2, \ldots, t_{N-1}\}$.
9:         Calculate the segment boundary $t_b$ using equation: $t_b = \left\lfloor \frac{t_n}{\lfloor \frac{T}{k} \rfloor} \right\rfloor \times \lfloor \frac{T}{k} \rfloor$.
10:        Sample ending timestep $t_{end}$ uniformly from $[t_b, t_{n-1}]$.
11:        Sample random noise $z$ from the normal distribution $\mathcal{N}(0, I)$.
12:        Sample the noised latent $x_{t_n} \sim \mathcal{N}(\alpha(t_n)z; \sigma^2(t_n)I)$.
13:        Compute the target $\hat{x}_{t_{n-1}}$ using Eq. (1).
14:        Compute the TSCD loss $L_{TSCD}$ using Eq. (12).
15:        Apply gradient descent to update $\Theta \leftarrow \Theta - \eta \nabla_\Theta L_{TSCD}$.
16:        Update the EMA of model parameters $\Theta^- \leftarrow \text{stopgrad}(\mu\Theta^- + (1 - \mu)\Theta)$.
17:     **end for**
18: **end for**
19: **Output:** Refined model parameters $\Theta$.

---

# C  Qualitative Results

## C.1  SD15 Architecture with LoRA training

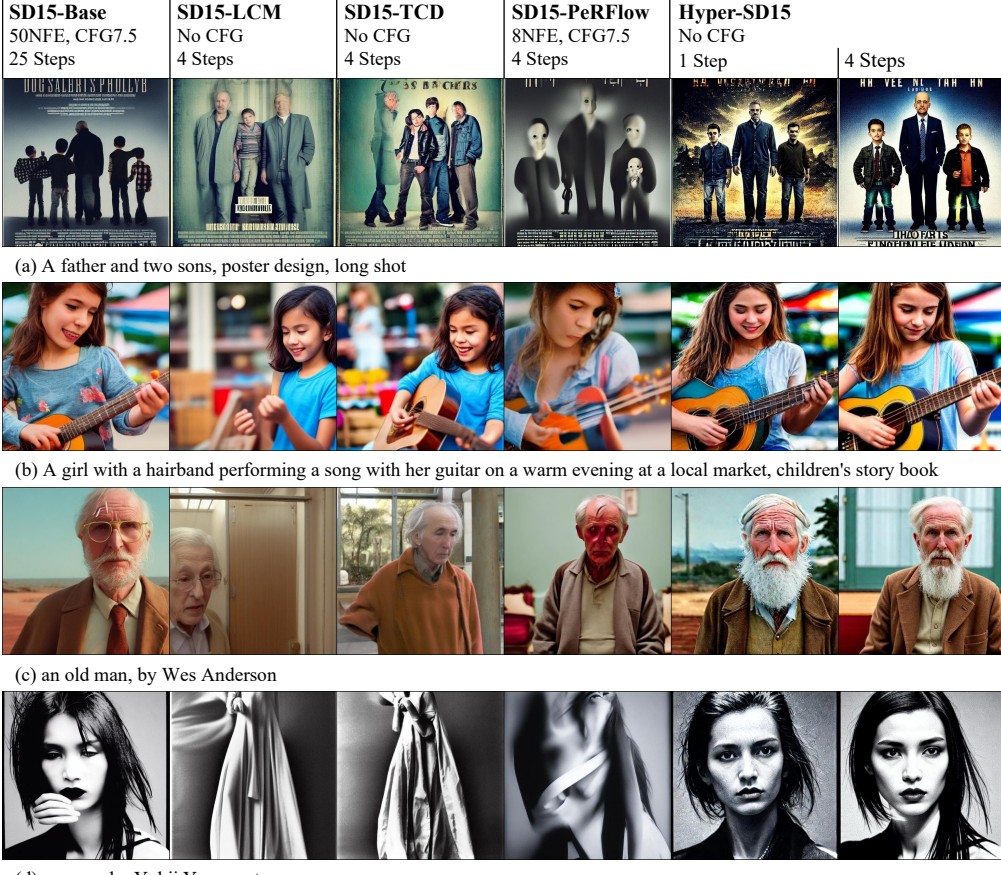

(a) A father and two sons, poster design, long shot

(b) A girl with a hairband performing a song with her guitar on a warm evening at a local market, children's story book

(c) an old man, by Wes Anderson

(d) woman, by Yohji Yamamoto

Figure 6. Qualitative comparisons with LoRA-based approaches on SD15 architecture.

## C.2  SDXL Architecture with UNet training

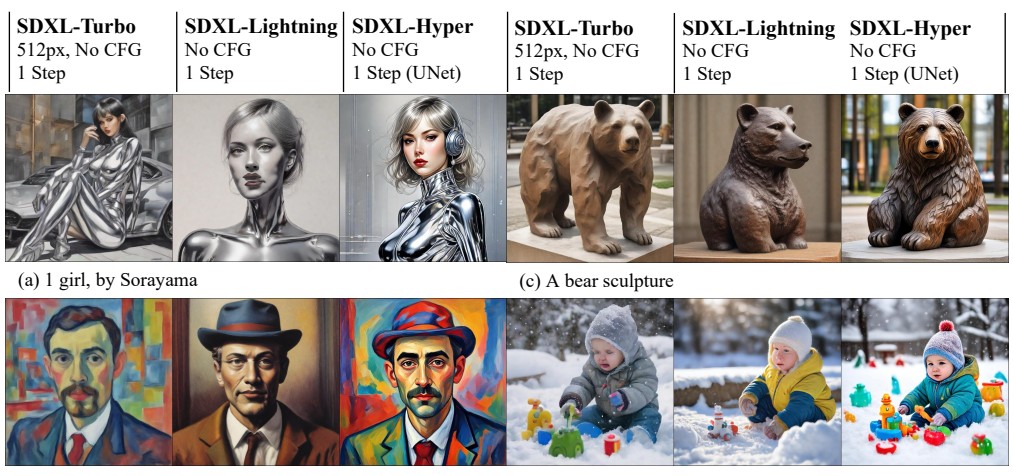

(a) 1 girl, by Sorayama

(c) A bear sculpture

(b) A portrait, Fauvist

(d) Baby playing with toys in the snow

Figure 7. Qualitative comparisons with UNet-based approaches on SDXL architecture.

## C.3 Unified LoRA

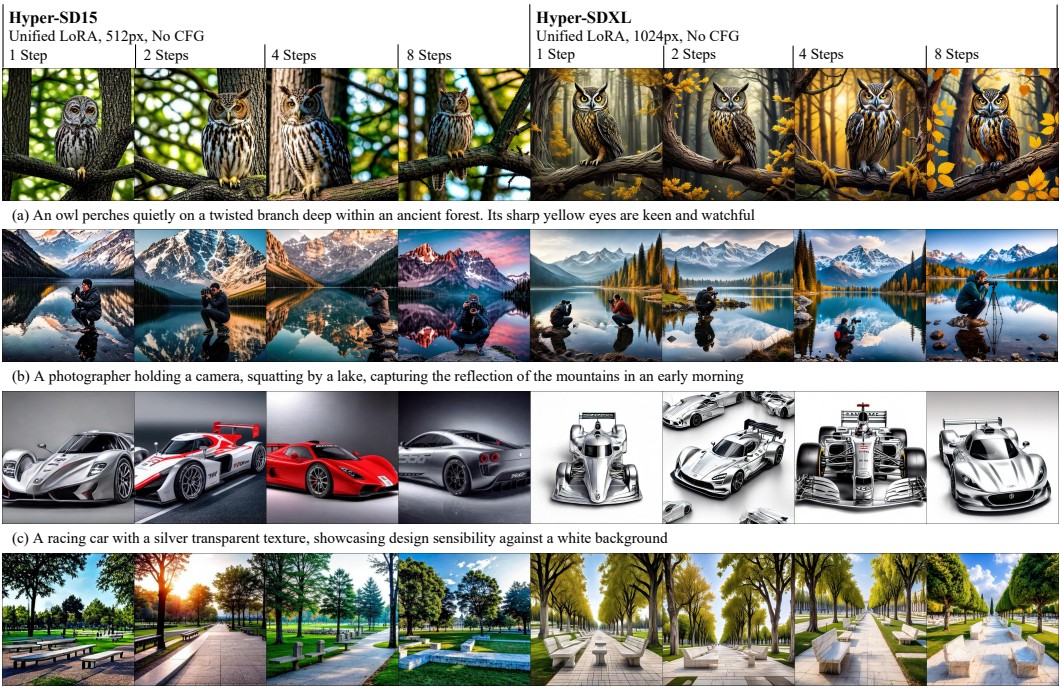

(a) An owl perches quietly on a twisted branch deep within an ancient forest. Its sharp yellow eyes are keen and watchful

(b) A photographer holding a camera, squatting by a lake, capturing the reflection of the mountains in an early morning

(c) A racing car with a silver transparent texture, showcasing design sensibility against a white background

(d) A tranquil park furnished with rows of benches made of marble

Figure 8. Qualitative results on unified LoRAs.

## C.4 Compatibility with ControlNet

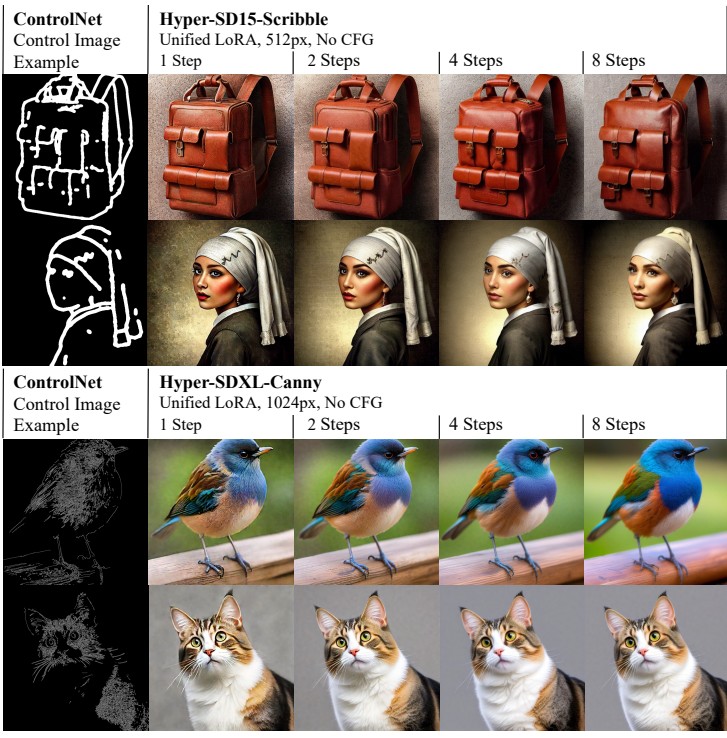

Figure 9. Our unified LoRAs are compatible with ControlNet. The examples are conditioned on either scribble or canny images.

## C.5    Compatibility with Base Model

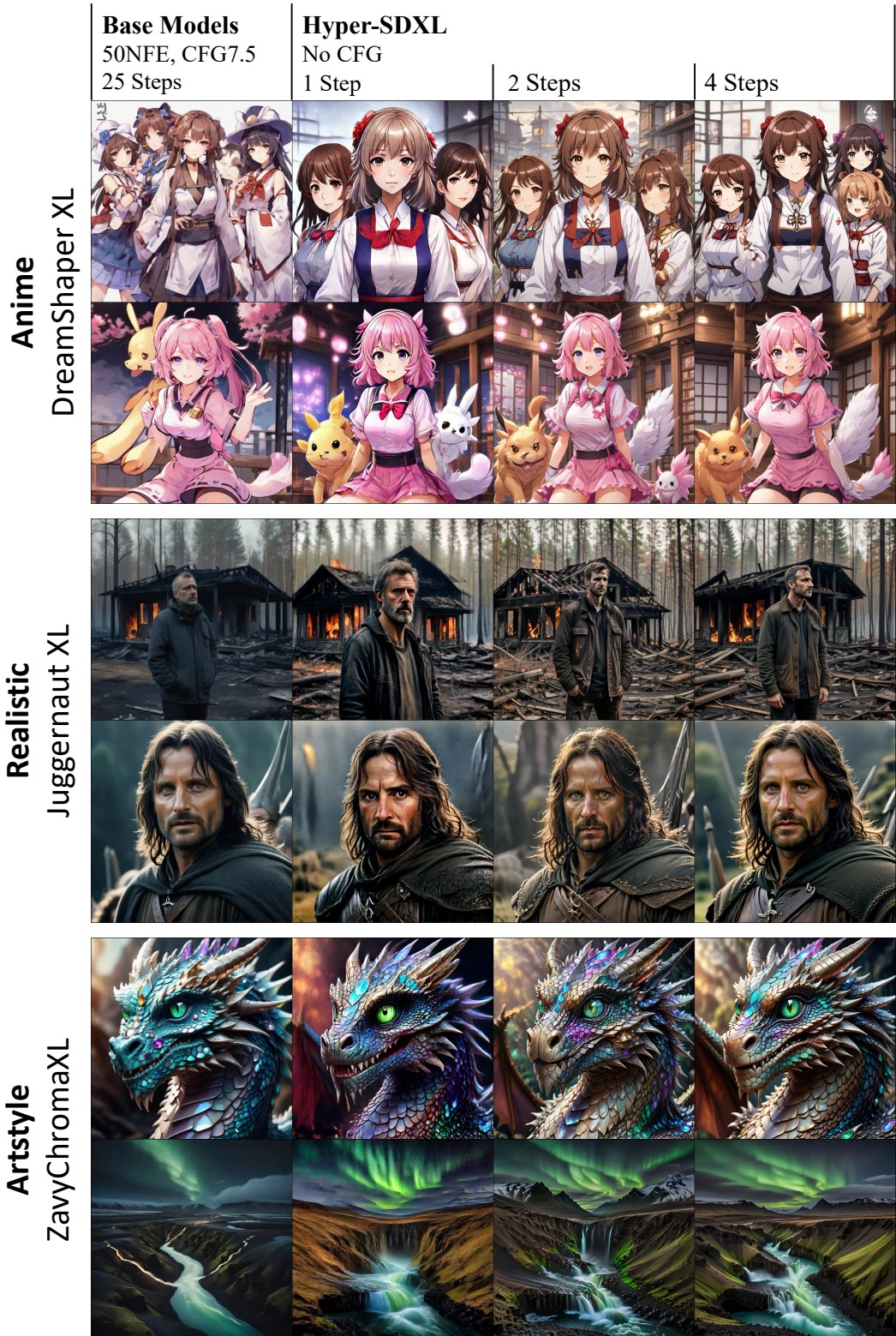

Figure 10. Our LoRAs with different steps can be applied to different base models and consistently generate high-quality images.

# D More ablation studies against TCD

To prove the effectiveness of TSCD against TCD, we conduct extra experiments on TCD+RLHF and TCD+DMD in Tab. 5. The results show that our TSCD demonstrates superior performance consistently over different training settings, which prove the robustness of TSCD with reduced training difficulty and less accumulation errors.

Table 5. More ablation studies against TCD.

| Method | Step | CLIP Score | Aes Score | Image Reward | Pick Score |
|---|---|---|---|---|---|
| DMD | 1 | 32.35 | 5.81 | 0.35 | 0.217 |
| TCD+DMD | 1 | 32.50 | 5.67 | 0.31 | 0.215 |
| **TSCD+DMD** | 1 | 32.58 (+0.08) | 5.69 (+0.02) | 0.85 (+0.54) | 0.222 (+0.007) |
| TCD | 2 | 32.36 | 5.62 | 0.29 | 0.217 |
| **TSCD** | 2 | 32.49 (+0.13) | 5.58 (-0.04) | 0.64 (+0.35) | 0.222 (+0.005) |
| TCD+RLHF | 2 | 32.38 | 5.63 | 0.59 | 0.221 |
| **TSCD+RLHF** | 2 | 32.61 (+0.23) | 5.84 (+0.21) | 1.04 (+0.45) | 0.232 (+0.011) |
| TCD | 4 | 32.45 | 5.42 | 0.67 | 0.226 |
| **TSCD** | 4 | 32.53 (+0.08) | 5.66 (+0.24) | 0.78 (+0.11) | 0.229 (+0.003) |
| TCD+RLHF | 4 | 32.50 | 5.62 | 0.85 | 0.229 |
| **TSCD+RLHF** | 4 | 32.56 (+0.06) | 5.74 (+0.12) | 0.93 (+0.08) | 0.232 (+0.003) |

# E Discussion and Limitation

Hyper-SD demonstrates promising results in generating high-quality images with few inference steps and could benefit various downstream tasks such as semi-supervised learning [43, 41], domain adaptation [31, 17], retrieval [42, 18], etc. However, there are several avenues for further improvement:

**Classifier Free Guidance**: the CFG properties of diffusion models allow for improving model performance and mitigating explicit content, such as pornography, by adjusting negative prompts. However, most diffusion acceleration methods [32, 46, 29, 14, 40, 28] including ours, eliminated the CFG characteristics, restricting the utilization of negative cues and imposing usability limitations. Therefore, in future work, we aim to retain the functionality of negative cues while accelerating the model, enhancing both generation effectiveness and security.

**Customized Human Feedback Optimization**: this work employed the generic reward models for feedback learning. Future work will focus on customized feedback learning strategies designed specifically for accelerated models to enhance their performance.

**Diffusion Transformer Architecture**: Recent studies have demonstrated the significant potential of DIT in image generation, we will focus on the DIT architecture to explore superior few-steps generative diffusion models in our future work.

