# OpenReview forum: "Hyper-SD: Trajectory Segmented Consistency Model for Efficient Image Synthesis"
_NeurIPS.cc/2024/Conference — NeurIPS 2024 poster_

### Official Review · Reviewer_5gi9 · 2024-06-26

**Soundness:** 2
**Presentation:** 2
**Contribution:** 2
**Rating:** 3
**Confidence:** 3

**Summary:**

This paper introduces Hyper-SD, a novel framework designed to mitigate the computational overhead associated with the multi-step inference process of Diffusion Models (DMs). The proposed framework addresses the limitations of existing distillation techniques, which focus on either ODE Trajectory Preservation or ODE Trajectory Reformulation and often suffer from performance degradation or domain shifts. Hyper-SD synergistically combines the advantages of both approaches while maintaining near-lossless performance during step compression. Key innovations include Trajectory Segmented Consistency Distillation for preserving the original ODE trajectory, human feedback learning to enhance low-step performance, and score distillation to improve low-step generation capabilities. The framework also uniquely leverages a unified LoRA for all inference steps. Extensive experiments and user studies demonstrate the Hyper-SD's supreme performance 1 to 8 inference steps for both SDXL and SD1.5, surpassing existing methods such as SDXL-Lightning in CLIP and Aes scores.

**Strengths:**

1. The paper comprehensively considers segmented distillation and the incorporation of human feedback scores, potentially outperforming the original diffusion models in certain cases. It even enables one-step image generation.
2. The experiments are thorough and well-executed and the paper is well-writen.

**Weaknesses:**

1. The additional introduction of the human feedback model might result in excessive memory usage. Since the evaluation steps require decoding and an additional model, would the actual computational resource consumption be clarified?
2. The inclusion of reward terms in the loss functions of LCM-based methods has been explored in related works, such as Reward Guided Latent Consistency Distillation. A comparison with these methods, if feasible, would be beneficial.

**Questions:**

refer to weakness

**Limitations:**

refer to weakness

---

> ### Author Rebuttal · Authors · 2024-08-05
>
> Thanks for your kind words about our method, experiments and writing. We would like to answer every question you have in the following.
>
> W1:
>
> We would like to clarify that our RLHF training is separated from TSCD and we apply RLHF through LoRA merging. So there would be no excessive memory usage in our distillation process. During RLHF, the convergence speed of RLHF LoRA is exceptionally rapid, requiring approximately 8000 iterations for around 96 A100 GPU hours. More importantly, the reward model is not involved in the evaluation process so no additional model needs to be decoded.
>
>
> W2:
>
> Thanks for your suggestion. We have read the provided paper carefully and will include it in our related work and discussion. The most essential difference is that we view feedback learning as a technique for trajectory-reformulating parallel to trajectory-preserving, and thus the training process is also separated. While in Reward Guided Latent Consistency Distillation (RG-LCM), the authors view it as some kind of guidance towards biasing the distillation process through a unified training objective.
>
> Our advantage is that we do not need to modify the original model's scheduler (i.e. DDIM) and thus have greater generalizability, while in RG-LCM the model needs to directly predict x_0 and fit the LCM scheduler. Moreover, our approach supports one-step generation with better trajectory-preserving and trajectory-reformulating pieces.

---

> ### Comment · Area_Chair_NzJv · 2024-08-10
> **please respond to authors**
>
> Hi,
>
> Please read other reviews and respond to the authors' rebuttal. You should edit your review accordingly.
>
> Thanks.

---

### Official Review · Reviewer_nBsQ · 2024-07-03

**Soundness:** 3
**Presentation:** 3
**Contribution:** 3
**Rating:** 7
**Confidence:** 5

**Summary:**

The paper introduces a novel framework named Hyper-SD, which enhances diffusion models’ efficiency by combining consistency distillation, human feedback learning, and distribution matching. Hyper-SD performs consistency distillation in segments to preserve the original ODE trajectory, maintaining high-quality generation while reducing inference steps. Human feedback learning further boosts performance in low-step inference, mitigating performance loss during the distillation process. Experimental results demonstrate that Hyper-SD achieves state-of-the-art performance from 1 to 8 inference steps, significantly surpassing existing methods, particularly in aesthetic scores and CLIP scores.

**Strengths:**

1. The components of this work are clear:
a) Based on TCD, the author segments the ODE and then performs distillation on these segments.
b) To enhance the model’s overall performance, the author utilizes RLHF techniques.
c) The author further employs the DMD scheme to optimize the results obtained in a single step.

2. The results are solid, clearly demonstrating the effectiveness of the proposed approach.

**Weaknesses:**

The primary issue is the insufficiency of the ablation experiments, the whole Hyper-SD sounds contain 2 modules, a TCSD LoRA, a RLHF LoRA, and TCSD LoRA for 1 step needs further DMD loss.   However, the ablation study does not demonstrate the effectiveness of each part. Here's my concerns:

1. In TCSD part, the distance metric includes GAN and L2. However, in TCD, it only contains L2. Is this metric similar to CTM (L2+GAN) or different? And how's the contribution of GAN part?

2. RLHF module needs another LoRA, is this the only different between Tab 3 w/ and w/ RLHF?

3. May I know the RECALL of the module w/ (w/o) GAN, w/ (w/o) RLHF ?

4. RLHF module is totally separately trained from the TCSD module?

5. The one-step model is the same training pipeline with TCSD (which means [8,4,2,1] + RLHF) but with DMD loss based on some base models like SDXL? Or it is a continue trained model based on a TCSD LoRA?

6. For one-step generator for comparison, does it include RLHF LoRA?

7. A minor question. Is $\Delta t$  ( or we can say $t_n - t_{n-1}$ ) the same for different segs?  My feeling is it should be same, but in Fig.2, the length of ODE solver is obviously different.

**Questions:**

Please check Weaknesses.

**Limitations:**

the limitations are discussed

---

> ### Author Rebuttal · Authors · 2024-08-05
>
> Thanks for your kind words about our method and main results. We would like to answer every question you have in the following.
>
> W0:
>
> We fully understand your primary concerns about ablation studies. To further validate the effectiveness of our proposed method in multi-step or one-step generation, we would like to raise your attention to Table 1 in the global rebuttal pdf, where we conduct experiments on TCD+RLHF, and TCD+DMD.
> 1. From the results, performance across different training settings when replacing the TSCD with TCD has decreased significantly.
> 2. As for RLHF component, we can see that vanilla TCD demonstrates severe aesthetic dropping when no RLHF LoRA is applied. In contrast, TCD+RLHF performs better but still lags behind our full Hyper-SD thanks to the refined trajectory-preserving capability from TSCD.
> 3. For one-step generation, the conclusions are similar that only after our TSCD performs better trajectory-preserving, the score estimation will be more accurate and the power of DMD to match teacher distribution will be stronger.
>
> W1:
>
> Our GAN implementation is different from CTM that adopts StyleGAN-XL[1] discriminator for unconditional or classifier-guidance image synthesis. Since the prompting in text-to-image diffusion is quite important, we turn to a more advanced discriminator from SDXL-Lightning[2] that uses U-Net for backbone and student model for initialization. Therefore, the inputs to the discriminator would be noisy latent and timesteps together and it could better capture the subtle variation between the prediction and targets across timesteps.
>
> To highlight the importance of introducing GAN loss, we conduct extra ablation studies in Table 3 of global rebuttal pdf. The results show that even if we also use adversarial loss for TCD, its performance is not as good as TSCD, which verifies our leadership in trajectory-preserving.
>
> W2:
>
> Yes it is. We apply RLHF through LoRA merging. It's exciting to see that simple merging of TSCD and RLHF LoRAs could bring significant benefits instead of distortion or collapse, which further demonstrates the importance of breaking down training objectives into trajectory-preserving and trajectory-reformulating pieces.
>
> W3:
>
> We have provided a more detailed comparison w/ or w/o RLHF (Table 1), w/ or w/o GAN (Table 3) in global rebuttal pdf. The results fully demonstrate the effectiveness of our proposed modules. For one, RLHF can still be effective even on TCD, and the results are even better on TSCD with better trajectory-preserving. For another, our GAN loss performs better against L2 only on TCD, and our TSCD benefit more from adversarial loss.
>
> W4:
>
> Not exactly. The training process of RLHF LoRA is with frozen TSCD LoRA applied. And we merge them into one after RLHF for convenience.
>
> W5:
>
> It is a continuously trained model based on the two-step TCSD LoRA since we set a progressive training strategy.
>
> W6:
>
> Yes it is. The result for DMD in Table 3 of manuscript is with RLHF. We're sorry to bring you this confusion. We have provided a clearer comparison for one-step generation w/o RLHF in Table 1 of global rebuttal pdf.
>
> W7:
>
> It's the same within a stage but different between stages. Like in the 4-segments stage, the Δt is 1000//16=62. While in the 2-segments stage, the Δt is 1000//8=125.
>
> > [1] Sauer, Axel, Katja Schwarz, and Andreas Geiger. "Stylegan-xl: Scaling stylegan to large diverse datasets." ACM SIGGRAPH 2022 conference proceedings. 2022.
> > [2] Lin, Shanchuan, Anran Wang, and Xiao Yang. "Sdxl-lightning: Progressive adversarial diffusion distillation." arXiv preprint arXiv:2402.13929 (2024).

---

> > ### Comment · Reviewer_nBsQ · 2024-08-11
> >
> > I have reviewed the author's response. Despite some issues in the experimental section, I believe the paper demonstrates sufficient innovation. The author's response has nearly resolved my concerns. I consider this paper to be slightly above the acceptance threshold. However, given that some reviewers have given it overly low scores, I am raising my score to 7 to balance my independent evaluation with the overall average score of the manuscript.

---

> ### Comment · Area_Chair_NzJv · 2024-08-10
> **respond to authors rebuttal**
>
> Hi,
>
> Please read other reviews and respond to the authors' rebuttal. You should edit your review accordingly.
>
> Thanks.

---

### Official Review · Reviewer_JxBk · 2024-07-13

**Soundness:** 3
**Presentation:** 3
**Contribution:** 2
**Rating:** 6
**Confidence:** 5

**Summary:**

This paper presents an approach for distilling a diffusion model into a multi-step generator. Previous distillation methods typically fall into two categories: those that preserve the ODE (Ordinary Differential Equation) trajectory and those that match the teacher model at the distribution level. This research offers a unified solution by integrating ideas from both categories. Specifically, it employs a multi-segment consistency distillation objective, akin to the ODE trajectory-preserving methods like Consistency Trajectory Model, and incorporates a combination of GAN (Generative Adversarial Network) and distribution matching distillation losses from the distribution-level matching methods. Additionally, an important contribution of this work is the incorporation of human or reward model feedback in the distillation process. The resulting method demonstrates high-quality text-to-image generation.

**Strengths:**

S1. The paper is well-written and easy to follow, providing a thorough introduction to the background and connections to previous works.

S2. The final approach is effective, with solid ablation studies of various components. The evaluations are comprehensive and demonstrate strong results.

S3. The utilization of reward optimization in diffusion distillation is relatively new, showcasing the potential of unifying distillation with other post-training techniques to enhance final performance in terms of aesthetics, alignment, and efficiency.

**Weaknesses:**

W1. The final method combines well-explored modules (consistency trajectory model, GAN, and DMD) into a unified framework. While this effective integration is a contribution, it is not particularly novel and doesn't introduce significantly new knowledge.

W2.Regarding the evaluation, diversity comparisons are missing from the current paper. Although FID is not the best metric, the authors should still report it for a comprehensive understanding. Additionally, assessing per-prompt diversity is important. The authors are encouraged to generate a set of images with the same prompt and measure their variations as a proxy metric. I am particularly interested in the influence of reward optimization on generation diversity.

W3. For one-step generation, the improvements appear small compared to vanilla DMD (Table 3).

**Questions:**

My concerns are outlined in the weaknesses section. While the paper demonstrates strong results and well-executed experiments, the overall method lacks significant novelty or performance improvement. Therefore, I will give it a borderline accept as my initial rating.

**Limitations:**

Yes

---

> ### Author Rebuttal · Authors · 2024-08-05
>
> Thanks for your kind words about our writing, experiments and reward optimization approaches. We would like to answer every question you have in the following.
>
> W1:
>
> As for technical novelty, we would like to highlight and summarize as follows:
> 1. We're the first to split the acceleration objective of diffusion model into trajectory-preserving and trajectory-reformulating. While previous approaches all just focus on the specific aspect, we break the objective down into pieces and yield different technical solutions, bringing new insights and perspectives in this field.
> 2.  As for trajectory-preserving, we're based on TCD[2] and hope to further detach the training objectives as neighboring timesteps and more distant timesteps to different training stages, which could reduce training difficulty and mitigate accumulation errors.
> 3. As for trajectory-reformulating, we propose to consider human feedback learning as an effective technique to bias output distribution toward human preference. It's exciting to see that simple merging of TSCD and RLHF LoRAs could bring significant benefits instead of distortion or collapse, which further demonstrates the importance of breaking down training objectives into pieces.
>
> W2:
>
> We agree that assessing per-prompt diversity is quite important and would like to raise your attention to Table 2 in the global rebuttal pdf, where we have picked 100 prompts and generated 4 images per prompt for each method shown. We extract the CLIP image embeddings per image and report their pair-wise average similarities for quantitative evaluation. The results show that our diversity without RLHF is quite similar compared to other acceleration approach. While biasing the output distribution toward human preference would inevitably compromise the diversity, our reward optimization doesn't sacrifice significantly and achieve an excellent trade-off compared to SDXL base model with a slight similarity increase (+0.0128). In Figure 2 of global rebuttal pdf, we also illustrate several examples for qualitative comparison.
>
> W3:
>
> We're sorry to bring you this confusion. The DMD result in Table 3 of manuscript is not vanilla but is with RLHF and without TSCD. To further validate the effectiveness of our proposed method in one-step generation, we have provided another ablation study in Table 1 of global rebuttal pdf. The results show that vanilla DMD doesn't yield competitive results compared to consistency distilled models in terms of CLIPScore, ImageReward and PickScore, while our TSCD performs to be better.

---

> > ### Comment · Reviewer_JxBk · 2024-08-07
> >
> > I thank the authors for their reply. I raise my score to 6. I agree with the authors about the innovation of integrating reward modeling in distillation. For diversity, it might be better to do something more pixel correlated e.g. LPIPS distance.

---

> > > ### Author Response · Authors · 2024-08-07
> > >
> > > Thanks again for your generous reply.
> > >
> > > We have just followed your advice on evaluating the generation diversity through pixel-level LPIPS metric.
> > >
> > > Using the same 100 prompts and random seed to generate 4 images each as in Table 3 of global rebuttal pdf, we list the average LPIPS distance as follows:
> > > - SDXL-Base (25-step UNet): 0.6991
> > > - SDXL-LCM (4-step LoRA): 0.6782
> > > - SDXL-TCD (4-step LoRA): 0.6668
> > > - SDXL-Lightning (4-step LoRA): 0.6895
> > > - TSCD (4-step LoRA): 0.7008
> > > - TSCD+RLHF (4-step LoRA): 0.6993
> > >
> > > Since higher LPIPS distance indicate better diversity, the results are quite to our surprise that our method surpass all the other acceleration approaches and even the baseline 25-step base model. And it is reasonable to see that with RLHF the LPIPS distance degrades a little bit.
> > >
> > > Benefiting from our better trajectory-preserving technique, we observe TSCD to have +0.034 LPIPS distance bonus against vanilla TCD. Since LPIPS is more interested in the subtle variations of color & detail, and the original 25-step base model is more towards gray in color style, our RLHF complements nicely to be more vivid so the metric doesn't compromise significantly (-0.0015).
> > >
> > > If you have further questions, please feel free to comment here. We welcome more discussion at any time.
> > >
> > > Thanks.

---

### Official Review · Reviewer_XyiH · 2024-07-17

**Soundness:** 3
**Presentation:** 3
**Contribution:** 2
**Rating:** 5
**Confidence:** 4

**Summary:**

This paper studies the distillation problem of diffusion models. Specifically, it introduces Trajectory Segmented Consistency Distillation to progressively perform consistent distillation within pre-defined time-step segments, which facilitates the preservation of the original ODE trajectory. Besides, the human feedback learning and Distribution Matching Distillation (DMD) technique are also included in the proposed model.

**Strengths:**

+ This paper is well written and easy to follow.

+ The studied problem is interesting and the proposed technique follows up the previous works to extend to a more general training strategy.

+ The user study is informative to demonstrate the superior performance of the proposed model.

**Weaknesses:**

- The proposed method of TSCD sounds interesting but some details may be questionable. First, intuitively, the randomness of time segments and the choices of t_end may introduce the randomness in the model training process. How can we guarantee the model will always converge to the same results given these different design choices? Second, comparing with CTM, how can we guarantee the multi-step training with different time steps can essentially introduce the advantages? For example, the multi-stage training may also introduce accumulated errors.

- It is interesting to add human feedback as a guidance. The paper also mentioned that human feedback learning may distort the output distribution. This is quite important. How can we know how human feedback learning is changing the distribution? How to detect and evaluate that? It is not very obvious why employing the LoRA merge technique with the TSCD can achieve a flexible balance between generation quality and output domain similarity. Can the author elaborate more on this? Any experiments or evidence to support this claim?

- Concern about the technical novelty. The proposed model consists of multiple techniques that have been introduced and commonly used in previous works. Although the performance looks promising by adding up all these modules together from the user study and the proposed method looks reasonable, there may not be some essential out-of-box ideas or findings that are very exciting.

- More detailed ablation study would be helpful. It is appreciated that the author conducts the ablation study to evaluate the effects of different modules. However, it is desirable to have a more thorough ablation study besides Table 3, since it may be still not very clear which modules among the three components play the most important factor in the proposed method. For example, without human feedback, the performance of TCD and TSCD is quite close, especially with more steps. How about with human feedback to TCD and then compare it? Does DMD are added in both TCD and TSCD experiments to have a fair comparison?

**Questions:**

Please see the weaknesses for specific questions.

**Limitations:**

The author discusses the limitations of the proposed method in the appendix, although it focuses more on the potential future works and follow up directions, instead of limitations and potential negative societal impact of their work.

---

> ### Author Rebuttal · Authors · 2024-08-05
>
> Thanks for your kind words about our writing, intuition and experiments. We would like to answer every question you have in the following.
>
> W1:
>
> Firstly, the randomness of t_end that broadened the boundary condition of the original consistency distillation in CM[1] (Theorem 1) has been proven to be valid and guaranteed to converge in TCD[2] (Theorem 4.1) or CTM[3] (Appendix B.1). And our time segments within each stage are fixed and identical, it just gets progressively smaller between stages.
>
> Secondly, comparing with CTM, our multi-stage design choice is actually breaking down the training objective into several pieces. During the early training stage, the model focuses on learning the consistency of nearby neighboring timesteps. While in the later stages, the model can easily handle those with small intervals and extend its capability to the consistency of further timesteps or larger intervals based on that. In other words, there is overlap in the training objectives between stages, even when there are fewer segments in the later stages, we still have a probability that the target timestep with a small interval will be stepped on.
>
> From the perspective of accumulation error, it comes from the difficulty of fitting the model from simple tasks (smaller intervals) to difficult tasks (larger intervals), and this is much smaller than the fitting error that comes from learning a mixed task (arbitrary intervals). Similar ideas could be evidenced by Progressive Distillation[4], where the promising distilled results indicate a smaller fitting error compared to the original non-distilled diffusion model. In our experiments, we also conduct extensive ablation and user studies to validate our superior performance against TCD across different settings and inference steps. For more ablation, we also conduct extra experiments in Table 1 of global rebuttal pdf which will be detailed later in W4 section.
>
> W2:
>
> Exactly, we would like to highlight that one of our main contributions is considering human feedback learning to be an effective technique to reformulate better trajectories, thus output distribution could be biased toward human preference rather than distorted badly. Since human preference is subjective and hard to be expressed through quantitative metrics, our assessment is also quite straightforward, that is, through extensive user studies as we show in the manuscript. To elaborate more on this, we would like to raise your attention to Figure 1 in the global rebuttal pdf, where we demonstrate the effectiveness of human feedback learning by different mixing ratios for TSCD and RLHF LoRAs.
>
> W3:
>
> As for technical contributions, we would like to highlight and summarize as follows:
> 1. We're the first to split the acceleration objective of diffusion model into trajectory-preserving and trajectory-reformulating. While previous approaches all just focus on the specific aspect, we break the objective down into pieces and yield different technical solutions, bringing new insights and perspectives in this field.
> 2.  As for trajectory-preserving, we're based on TCD[2] and hope to further detach the training objectives as neighboring timesteps and more distant timesteps to different training stages, which could reduce training difficulty and mitigate accumulation errors.
> 3. As for trajectory-reformulating, we propose to consider human feedback learning as an effective technique to bias output distribution toward human preference. It's exciting to see that simple merging of TSCD and RLHF LoRAs could bring significant benefits instead of distortion or collapse, which further demonstrates the importance of breaking down training objectives into pieces.
>
> W4:
>
> We fully understand your concerns about ablation studies. To prove the effectiveness of TSCD against TCD, we would like to raise your attention to Table 1 in the global rebuttal pdf, where we conduct experiments on TCD+RLHF and TCD+DMD. The results show that our TSCD demonstrates superior performance consistently over different training settings, which prove the robustness of TSCD with reduced training difficulty and less accumulation errors.
>
> > [1] Song, Yang, et al. "Consistency Models." International Conference on Machine Learning. PMLR, 2023.
> > [2] Zheng, Jianbin, et al. "Trajectory consistency distillation." arXiv preprint arXiv:2402.19159 (2024).
> > [3] Kim, Dongjun, et al. "Consistency Trajectory Models: Learning Probability Flow ODE Trajectory of Diffusion." The Twelfth International Conference on Learning Representations.
> > [4] Salimans, Tim, and Jonathan Ho. "Progressive Distillation for Fast Sampling of Diffusion Models." International Conference on Learning Representations.

---

> > ### Comment · Reviewer_XyiH · 2024-08-13
> >
> > Thanks for the author's efforts to answer my questions. Some of my concerns have been well addressed. I would like to raise my score to borderline accept. Regarding W2, it is great to see how the RLHF weighting factor is influencing the generated images. But without a thorough and rigors study, it is still hard to conclude how the distribution manifold is changed from these few samples, or what kind of bias is introduced. This may need to further concerns when this proposed model is used to other subsequent applications.

---

> ### Comment · Area_Chair_NzJv · 2024-08-10
> **Response to rebuttal.**
>
> Hi,
>
> Please read other reviews and respond to the authors' rebuttal. You should edit your review accordingly.
>
> Thanks.

---

### Author Rebuttal · Authors · 2024-08-06

Dear all,

For each question from different reviewers, we have responded individually with a targeted rebuttal under.

We put all the figures and tables into the pdf file submitted here.

Hope this address your concerns and we welcome more discussion at any time.

Regardless of the final decision, thank you all for your hard work.

Best,

Authors of Submission 2694.

---

### Decision · Program_Chairs · 2024-09-25

**Decision:**

Accept (poster)

**Comment:**

This paper has initially received borderline reviews. Though after the discussion period, most reviewers' concerns were answered and they have all increased their score.

One reviewer who most negatively judged the current submission did not participate in the discussion despite many reminders. In addition, his remarks come from a misunderstanding that has been answered by the authors. Therefore I decided to not take this review into account.

Other reviewers acknowledged the value and originality of the proposed work:
- "results are solid, clearly demonstrating the effectiveness of the proposed approach."
- "The final approach is effective, with solid ablation studies of various components. The evaluations are comprehensive and demonstrate strong results."
- "utilization of reward optimization in diffusion distillation is relatively new"
- "paper is well written and easy to follow."

The authors should update their paper according their discussion with the reviewers for the camera-ready version.